# FEW-SHOT FEW-SHOT LEARNING AND THE ROLE OF SPATIAL ATTENTION

## ABSTRACT

Few-shot learning is often motivated by the ability of humans to learn new tasks from few examples. However, standard few-shot classification benchmarks assume that the representation is learned on a limited amount of base class data, ignoring the amount of prior knowledge that a human may have accumulated before learning new tasks. At the same time, even if a powerful representation is available, it may happen in some domain that base class data are limited or non-existent. This motivates us to study a problem where the representation is obtained from a classifier pre-trained on a large-scale dataset of a different domain, assuming no access to its training process, while the base class data are limited to few examples per class and their role is to adapt the representation to the domain at hand rather than learn from scratch. We adapt the representation in two stages, namely on the few base class data if available and on the even fewer data of new tasks. In doing so, we obtain from the pre-trained classifier a *spatial attention map* that allows focusing on objects and suppressing background clutter. This is important in the new problem, because when base class data are few, the network cannot learn where to focus implicitly. We also show that a pre-trained network may be easily adapted to novel classes, without meta-learning.

## 1 INTRODUCTION

The ever improving performance of deep learning models, apart from technical progress, is largely due to the existence of large-scale datasets, fully or weakly labeled by humans (Russakovsky et al., 2015; Kuznetsova et al., 2018; Mahajan et al., 2018). At the same time, reducing the need for supervision is becoming increasingly important, *e.g.* by taking advantage of prior learning (Bengio et al., 2011; Mallya et al., 2018; Li & Hoiem, 2018) or exploiting unlabeled data (Caron et al., 2018; Radosavovic et al., 2018).

An extreme situation is *few-shot learning* (Koch et al., 2015; Vinyals et al., 2016; Snell et al., 2017; Hariharan & Girshick, 2017), where the problem is to learn *novel* (previously unseen) *classes* using only a very small labeled training set, typically not more than 10 examples per class. Here not only the annotation but even the raw data are not available. This problem is often motivated by the ability of humans to learn new tasks from few examples (Lake et al., 2011; 2015), which has given rise to *meta-learning* (Santoro et al., 2016; Mishra et al., 2017; Ravi & Larochelle, 2017; Bertinetto et al., 2018; Han et al., 2018), or learning to learn. In this scenario, a training set is treated as a collection of smaller sets where every class has few labeled examples.

However, there is a huge gap between the motivating example of humans learning new tasks and how the few-shot classification task is set up. On one hand, for the sake of simplicity in experiments, the *base class* datasets where the representation is learned from scratch, contain a few dozen or hundred classes with a few hundred examples each. This is by no means comparable to datasets available to date (Kuznetsova et al., 2018; Mahajan et al., 2018), let alone the amount of prior knowledge that a human may have accumulated before learning a new task. On the other hand, for a given domain of novel classes, *e.g.* bird species (Welinder et al., 2010), base class data of such size in the same domain may not exist.

In this work, we depart from the standard few-shot classification scenario in two directions. First, we allow the representation to be learned from a large-scale dataset in a domain different than the base and novel class domain. In particular, we model prior knowledge by a classifier that is *pre-trained* on

such a dataset, having no access to its training process. We thus maintain the difficulty of domain gap and the simplicity of experiments (by not training from scratch), while allowing a powerful representation. Second, we assume only *few or zero* examples per base class. Hence, the role of base classes is to *adapt* the representation to the domain at hand rather than learn from scratch. This scenario can be seen as few-shot version of few-shot learning.

We treat this problem as a two-stage adaptation process, first on the few base class examples if available and second on the even fewer novel class examples. Because of the limited amount of data, it is not appropriate to apply *e.g. transfer learning* (Bengio et al., 2011) or *domain adaptation* (Ganin & Lempitsky, 2014), in either of the two stages. Because the network is pre-trained, and we do not have access to its training process or data, *meta-learning* is not an option either. We thus resort to few steps of fine-tuning as in the *meta-testing* stage of Finn et al. (2017) and Ravi & Larochelle (2017).

Focusing on image classification, we then investigate the role of *spatial attention* in the new problem. With large base class datasets, the network can implicitly learn the relevant parts of the images where to focus. In our setup, base class data are few, so our motivation is that a spatial attention mechanism may help the classifier in focusing on objects, suppressing background clutter. We observe that although the *prior classes* of the pre-trained classifier may be irrelevant to a new task, uncertainty over a large number of such classes may express anything unknown like background. This is a *class-agnostic* property and can apply to new tasks.

In particular, given an input image, we measure the entropy-based *certainty* of the pre-trained classifier in its prediction on the prior classes at every spatial location and we use it to construct a spatial attention map. This map can be utilized in a variety of ways, for instance weighted spatial pooling or weighted loss per location; and at different situations like the two adaptation stages or at inference. By exploring different alternatives, we show that a pre-trained network may be easily adapted to novel classes, without meta-learning.

In the following, we provide a detailed problem formulation and related background in section 2, then we describe our spatial attention mechanism in section 3 and its use in few-shot classification in section 4. We provide experimental results in section 5, and we conclude in section 6.

## 2 PROBLEM, BACKGROUND, RELATED WORK AND CONTRIBUTION

**Few-shot classification.** We are given a set of *training examples* $X := \{\mathbf{x}_i\}_{i=1}^n \subset \mathcal{X}$, and corresponding labels $\mathbf{y} := (y_i)_{i=1}^n \subset C^n$ where $C := [c] := \{1, \ldots, c\}$ is a set of *base classes*. The objective is to learn a representation on these data, a process that we call *base training*, such that we can solve new tasks. A new task comprises a set of *support examples* $X' := \{\mathbf{x}'_i\}_{i=1}^{n'} \subset \mathcal{X}$ and labels $\mathbf{y}' := (y'_i)_{i=1}^{n'} \subset (C')^{n'}$, where $n' \ll n$ and $C' := [c']$ is a set of *novel classes* disjoint from $C$. The most common setting is $k'$ examples per novel class, so that $n' = k'c'$, referred to as $c'$-*way*, $k'$-*shot* classification. The objective now is to learn a classifier on these support data, a process that we call *adaptation*. This classifier should map a new *query example* from $\mathcal{X}$ to a prediction in $C'$.

**Few-shot few-shot classification.** Few-shot classification assumes there is more data in base than novel classes, and a domain shift between the two, in the sense of no class overlap. Here we consider a modified problem where $n$ can be small or *zero*, but there is another set $C^\circ = [c^\circ]$ of *prior classes* with even more data $X^\circ$ and labels $\mathbf{y}^\circ$ with $n \ll n^\circ := |X^\circ|$ and a greater domain shift to $C, C'$. Again, the most common setting is $k$ examples per base class, so that $n = kc$. We are using a classifier that is *pre-trained* on this data but we do not have direct access to either $X^\circ, \mathbf{y}^\circ$, or its learning process. The objective of base training is now to adapt the representation to the domain of $C, C'$ rather than learn it from scratch; but we still call it *base training*.

In the remaining of this section we present general background on few-shot classification that typically applies to base classes $C$ or novel classes $C'$, but may also apply to both, in which case the symbols $c, C$ and $c', C'$ may be used interchangeably. Prior classes and the pre-trained network are only considered in the following sections.

**Classifier.** The *classifier* is a function $f_{\theta,W} : \mathcal{X} \to \mathbb{R}^c$ (resp. $\mathbb{R}^{c'}$) with learnable parameters $\theta, W$, mapping a new example $\mathbf{x} \in \mathcal{X}$ to a vector of probabilities $\mathbf{p} := f_{\theta,W}(\mathbf{x})$ over $c$ (resp. $c'$) base (resp.

novel) classes. The classifier *prediction* is the class of maximum probability

$$\pi(\mathbf{p}) := \arg\max_j p_j, \tag{1}$$

where $p_j$ is the $j$-th element of $\mathbf{p}$. The classifier is built on top of an *embedding function* $\phi_\theta : \mathcal{X} \to \mathbb{R}^{r \times d}$. Given an example $\mathbf{x} \in \mathcal{X}$, this function yields a $r \times d$ feature tensor $\phi_\theta(\mathbf{x})$, where $r$ represents the spatial dimensions and $d$ the feature dimensions. For $\mathcal{X}$ comprising 2d images for instance, the feature is a $w \times h \times d$ tensor that is the activation of the last convolutional layer, where $r = w \times h$ is the spatial resolution. The embedding is a vector in $\mathbb{R}^d$ in the special case $r = 1$.

The embedding parameters $\theta$ may be updated into $\theta'$ in the adaptation process, in which case we have an embedding function $\phi_{\theta'}$ and classifier $f_{\theta',W}$. Again $\theta, \theta'$ may be used interchangeably.

**Cosine classifier.** A simple form of classifier that was introduced in few-shot learning independently by Qi et al. (2018) and Gidaris & Komodakis (2018) is a parametric linear classifier that consists of a fully-connected layer without bias on top of the embedding function $\phi_\theta$ followed by softmax. If $W := (\mathbf{w}_j)_{j=1}^c$ is the collection of class weights with $\mathbf{w}_j \in \mathbb{R}^{r \times d}$, the classifier is defined by

$$f_{\theta,W}(\mathbf{x}) := \boldsymbol{\sigma}\left(\tau[s(\phi_\theta(\mathbf{x}), \mathbf{w}_j)]_{j=1}^c\right) \tag{2}$$

for $\mathbf{x} \in \mathcal{X}$, where $\boldsymbol{\sigma} : \mathbb{R}^m \to \mathbb{R}^m$ is the *softmax function* $\boldsymbol{\sigma}(\mathbf{u}) := \frac{(e^{u_1}, \dots, e^{u_c})}{\sum_j e^{u_j}}$ for $\mathbf{u} \in \mathbb{R}^c$, $\tau \in \mathbb{R}^+$ is a trainable *scale parameter* and $s$ is *cosine similarity*[1]. We minimize the *cost function*

$$J(X, \mathbf{y}; \theta, W) := \sum_{i=1}^n \ell(f_{\theta,W}(\mathbf{x}_i), y_i) \tag{3}$$

over $\theta, W$ at base training, where $\ell(\mathbf{p}, y) := -\log p_y$ for $\mathbf{p} \in \mathbb{R}_+^c$, $y \in C$ is the *cross-entropy* loss.

**Prototypes.** An alternative classifier that is more appropriate during few-shot adaptation or at testing is a *prototype classifier* proposed by Snell et al. (2017) and followed by Qi et al. (2018) and Gidaris & Komodakis (2018) too. If $S_j := \{i \in [n'] : y_i' = j\}$ denotes the indices of support examples labeled in class $j$, then the *prototype* of this class $j$ is given by the average features

$$\mathbf{p}_j = \frac{1}{|S_j|} \sum_{i \in S_j} \phi_{\theta'}(\mathbf{x}_i') \tag{4}$$

of those examples for $j \in C'$. Then, denoting by $P := (\mathbf{p}_j)_{j=1}^{c'}$ the collection of prototypes, a query $\mathbf{x} \in \mathcal{X}$ is classified as $f_{\theta',P}(\mathbf{x})$, as defined by (2).

**Dense classifier.** This is a classifier where the loss function applies densely at each spatial location of the feature tensor rather than by global pooling or flattening as implied by (2) . The classifier can be of any form but a cosine classifier (Qi et al., 2018; Gidaris & Komodakis, 2018) is studied by Lifchitz et al. (2019). In particular, the embedding $\phi_\theta(\mathbf{x})$ is seen as a collection of vectors $[\phi^{(q)}(\mathbf{x})]_{q=1}^r$, where $\phi^{(q)}(\mathbf{x}) \in \mathbb{R}^d$ is an embedding of spatial location $q \in [r]$. The classifier (2) is then generalized to $f_{\theta,W} : \mathcal{X} \to \mathbb{R}^{r \times c}$, now mapping an example to a vector of probabilities per location, defined by

$$f_{\theta,W}(\mathbf{x}) := \left[\boldsymbol{\sigma}\left(\tau[s(\phi_\theta^{(q)}(\mathbf{x}), \mathbf{w}_j)]_{j=1}^c\right)\right]_{q=1}^r \tag{5}$$

for $\mathbf{x} \in \mathcal{X}$, while the class weights $W$ are shared over locations with $\mathbf{w}_j \in \mathbb{R}^d$. Cross-entropy applies using the same label $y_i$ for each location of example $\mathbf{x}_i$, generalizing the cost function (3) to

$$J(X, \mathbf{y}; \theta, W) := \sum_{i=1}^n \sum_{q=1}^r \ell(f_{\theta,W}^{(q)}(\mathbf{x}_i), y_i). \tag{6}$$

**Related work.** *Prototypical networks* (Snell et al., 2017) use a prototype classifier. At testing, a query is classified to the nearest prototype, while at adaptation, computing a prototype per class (4)

---

[1] For matrices $\mathbf{u}, \mathbf{v} \in \mathbb{R}^{r \times d}$, $s(\mathbf{u}, \mathbf{v}) := \langle \mathbf{u}, \mathbf{v} \rangle / (\|\mathbf{u}\| \|\mathbf{v}\|)$ with $\langle \cdot, \cdot \rangle$ and $\|\cdot\|$ being the Frobenius inner product and norm respectively.

is the only learning to be done. Base training is based on *meta-learning* : a number of fictitious tasks called *episodes* are generated by randomly sampling a number of classes from $C$ and then a number of examples in each class from $X$ with their labels from $\mathbf{y}$. These data are assumed to be support examples and queries of novel classes $C'$. Labels are now available for the queries and the objective is that they are classified correctly. In *imprinting* (Qi et al., 2018), a cosine classifier (2) and standard cross-entropy (3) are used instead at base training. At adaptation, class prototypes $P$ are computed (4) and *imprinted* in the classifier, that is, $W$ is replaced by $W' := (W, P)$. The entire embedding function is then fine-tuned based again on (3) to make predictions on $n + n'$ base and novel classes, which requires the entire training data $(X, \mathbf{y})$. *Few-shot learning without forgetting* (Gidaris & Komodakis, 2018) uses model similar to imprinting , the main difference being that only the weight parameters $W$ of the base classes are stored and not the entire training data. At base training, a cosine classifier is trained by (3) followed by episodes. At adaptation, prototypes are adapted to $W$ by a *class attention* mechanism. *Model agnostic meta-learning* (MAML) (Finn et al., 2017) uses a fully-connected layer as classifier. At adaptation, the entire embedding function is fine-tuned on each new task using (3) only on the novel class data, but for *few steps* such that the classifier does not overfit. At base training, episodes are used where the loss function mimics the iterative optimization that normally takes place at adaptation. In *implanting* (Lifchitz et al., 2019), a prototype classifier is trained in episodes, keeping the base embedding function fixed but attaching a parallel *implant* stream of convolutional layers that learns features useful for each new task.

**Contribution.** The problem we consider is a variant of few-shot learning that has not been studied before. It involves sequential adaptation of a given network in two stages, each comprising a limited amount of data. There are many ways of exploiting prior learning to reduce the required amount data and supervision like *transfer learning* (Bengio et al., 2011; Yosinski et al., 2014), *domain adaptation* (Ganin & Lempitsky, 2014; Rebuffi et al., 2017; Mallya et al., 2018), or *incremental learning* (Li & Hoiem, 2018; Rebuffi et al., 2016; Yoon et al., 2018). However, none applies to the few-shot domain where just a handful of examples are given.

*Attention* has been studied as a core component of several few-shot learning and meta-learning approaches (Vinyals et al., 2016; Mishra et al., 2018; Ren et al., 2018; Gidaris & Komodakis, 2018), but it always refers to *examples* (*e.g.* images) as a unit. *Spatial attention* on the other hand refers to neurons at different spatial locations. It is ubiquitous in several problems, for instance weakly supervised object detection (Zhou et al., 2016; Hou et al., 2018; Zhu et al., 2017) and non-local convolution (Hu et al., 2018; Wang et al., 2018; Chen et al., 2018) but has not been applied to few-shot learning until recently (Lifchitz et al., 2019; Wertheimer & Hariharan, 2019; Li et al., 2019b; Liu et al., 2019). By breaking up one example (*e.g.* image) to several, one at each spatial location, the supervision becomes *weak*, as in *multiple-instance learning* (Andrews et al., 2003), typical *e.g.* in weakly-supervised image segmentation (Jetley et al., 2018). Our spatial attention mechanism is extremely simple, based on the pre-trained network, without training on the base or novel class data.

## 3 SPATIAL ATTENTION FROM PRE-TRAINING

We assume a pre-trained network with an embedding function $\phi_{\theta^\circ} : \mathcal{X} \to \mathbb{R}^{r \times d}$ followed by *global average pooling* (GAP) and a classifier that is a fully connected layer with weights $W^\circ := (\mathbf{w}_j^\circ)_{j=1}^{c^\circ} \in \mathbb{R}^{d \times c^\circ}$ and biases $\mathbf{b}^\circ \in \mathbb{R}^{c^\circ}$, denoted jointly by $U^\circ := (W^\circ, \mathbf{b}^\circ)$. Without re-training, we remove the last pooling layer and apply the classifier densely as in $1 \times 1$ convolution, followed by softmax with temperature $T$. Then, similarly to (5), the classifier $f_{\theta^\circ, U^\circ} : \mathcal{X} \to \mathbb{R}^{r \times c^\circ}$ maps an example to a vector of probabilities per location, where classifier parameters $U^\circ$ are shared over locations:

$$f_{\theta^\circ, U^\circ}(\mathbf{x}) := \left[ \boldsymbol{\sigma} \left( \frac{1}{T} \left( W^{\circ\top} \phi_{\theta^\circ}^{(q)}(\mathbf{x}) + \mathbf{b}^\circ \right) \right) \right]_{q=1}^{r}. \tag{7}$$

We now want to apply this classifier to examples in set $X$ (resp. $X'$) of base (resp. novel) classes $C$ (resp. $C'$) in order to provide a *spatial attention* mechanism to embeddings obtained by parameters $\theta$ (resp. $\theta'$). We formulate the idea on $X, C, \theta$ in this section but it applies equally to $X', C', \theta'$. In particular, given an example $\mathbf{x} \in X$, we use the vector of probabilities $\mathbf{p}^{(q)} := f_{\theta^\circ, U^\circ}^{(q)}(\mathbf{x})$ corresponding to spatial location $q \in [r]$ to compute a scalar weight $w^{(q)}(\mathbf{x})$, expressing the *discriminative power* of the particular location $q$ of example $\mathbf{x}$.

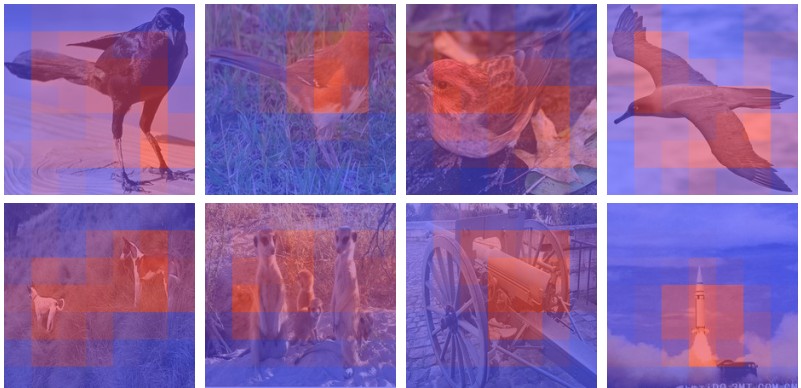

Figure 1: Examples of images from CUB (top) and *mini*ImageNet (bottom) overlaid with entropy-based spatial attention maps obtained from (8) using only the predicted class probabilites from ResNet-18 pre-trained on Places. See section 5 for details on datasets and networks.

Since $\mathbf{x}$ belongs to a set of classes $C$ different than $C^\circ$, there is no ground truth to be applied to the output of the pre-trained classifier $f_{\theta^\circ, U^\circ}$. However, the distribution $\mathbf{p}^{(q)}$ can still be used to evaluate how discriminative the input is. We use the entropy function for this purpose, $H(\mathbf{p}) := -\sum_j p_j \log(p_j)$. We map the entropy to $[0, 1]$, measuring the *certainty* of the pre-trained classifier in its prediction on the prior classes $C^\circ$:

$$w^{(q)}(\mathbf{x}) := 1 - \frac{H(f^{(q)}_{\theta^\circ, U^\circ}(\mathbf{x}))}{\log c^\circ} \tag{8}$$

for $q \in [r]$, where we ignore dependence on parameters $\theta^\circ, U^\circ$ to simplify notation, since they remain fixed. We use this as a weight for location $q$ assuming that uncertainty over a large number of prior classes expresses anything unknown like background, which can apply to a new set of classes. We then $\ell_1$-normalize the weights $w(\mathbf{x}) := [w^{(q)}(\mathbf{x})]^r_{q=1} \in \mathbb{R}^r$ as $\hat{w}(\mathbf{x}) := w(\mathbf{x})/\|w(\mathbf{x})\|_1$. We call $\hat{w}(\mathbf{x})$ the *spatial attention weights* of $\mathbf{x}$.

The weights are applied in different ways depending on the problem. If the embedding $\phi_\theta(\mathbf{x})$ is normally a vector in $\mathbb{R}^d$ obtained by GAP on a feature tensor $\Phi_\theta(\mathbf{x}) \in \mathbb{R}^{r \times d}$ as $\frac{1}{r} \sum_{q \in [r]} \Phi^{(q)}_\theta(\mathbf{x})$ for $\mathbf{x} \in \mathcal{X}$, then GAP is replaced by global global *weighted* average pooling (GwAP):

$$\phi_\theta(\mathbf{x}) := \sum_{q \in [r]} \hat{w}^{(q)}(\mathbf{x}) \Phi^{(q)}_\theta(\mathbf{x}). \tag{9}$$

for $\mathbf{x} \in \mathcal{X}$. We recall that this applies equally to $\theta'$ in the case of novel classes.

Figure 1 shows examples of images with spatial attention maps. Despite the fact that there has been no training involved for the estimation of attention on the particular datasets, the result can still be useful in suppressing background clutter.

## 4 SPATIAL ATTENTION IN FEW-SHOT CLASSIFICATION

Here we discuss the use of attention maps at inference on novel classes, as well as learning on novel classes. In the latter case, the weights are pre-computed for all training examples since the pre-trained network remains fixed in this process. In summary, we either replace GAP by GwAP (9) in all inputs to the embedding network, or use dense classification (5).

### 4.1 BASE CLASS TRAINING

Starting from a pre-trained embedding network $\phi_{\theta^\circ}$, we can either solve new tasks on novel classes $C'$ directly, in which case $\theta = \theta^\circ$, or perform base class training, fine-tuning $\theta$ from $\theta^\circ$. Adaptation may involve for instance fine-tuning the last layers or the entire network, applying a spatial attention

mechanism or not. Recalling that $\phi_{\theta^\circ}$ is still needed for weight estimation (8), the most practical setting is to fine-tune the last layers, in which case $\phi_\theta$ shares the same backbone network with $\phi_{\theta^\circ}$. Following MAML (Finn et al., 2017), we perform few gradient descent steps with low learning rate.

We use a dense classifier $f_{\theta,W} : \mathcal{X} \to \mathbb{R}^{r \times c}$ (5) with class weights $W$. Given the few base class examples $X$ and labels $\mathbf{y}$, we learn $W$ at the same time as fine-tuning $\theta$ by minimizing (6).

## 4.2 NOVEL CLASS ADAPTATION

Optionally, given the few novel class support examples $X'$ and labels $\mathbf{y}'$, we can further adapt the embedding network, while applying our attention mechanism to the loss function. As in section 4.1, $\phi_{\theta'}$ shares the same backbone with $\phi_\theta$, being derived from it by fine-tuning the last layers. We perform even fewer gradient descent steps with lower learning rate

We use a prototype classifier where vector embeddings $\phi_{\theta'}(\mathbf{x}')$ of support examples $\mathbf{x}' \in X'$ are obtained by GwAP with $\phi_{\theta'}$ defined as in (9) and class prototypes $P := (\mathbf{p}_j)_{j=1}^{c'}$ are obtained per class by averaging embeddings of support examples as defined by (4) and updated whenever $\theta'$ is updated. The classifier $f_{\theta',P} : \mathcal{X} \to \mathbb{R}^{c'}$ is a standard cosine classifier (2) and the loss function is standard cross-entropy $J(X, \mathbf{y}; \theta, P)$ (3) with embedding $\phi_{\theta'}$ obtained by GwAP (9). Attention weights apply to embeddings of all inputs to the network, each time focusing on most discriminative parts. In case of no adaptation to the embedding network, we fix $\theta' = \theta$. Computing the prototypes $P$ (4) is then the only learning to be done and we can proceed to inference directly.

## 4.3 NOVEL CLASS INFERENCE

At inference, as in section 4.2, we adopt a prototype classifier where vector embeddings $\phi_{\theta'}(\mathbf{x}')$ of support examples $\mathbf{x}' \in X'$ are obtained by GwAP with $\phi_{\theta'}$ defined as in (9) and class prototypes $P := (\mathbf{p}_j)_{j=1}^{c'}$ are obtained per class by averaging embeddings of support examples as defined by (4). Then, given a query $\mathbf{x} \in \mathcal{X}$, we similarly obtain a vector embedding $\phi_{\theta'}(\mathbf{x})$ by GwAP (9) and predict the class $\pi(f_{\theta',P}(\mathbf{x}))$ of the nearest prototype according to cosine similarity where $\pi$ is given by (1) and $f_{\theta',P}$ by (2). We thus focus on discriminative parts of both support and query examples, suppressing background clutter.

## 5 EXPERIMENTS

### 5.1 EXPERIMENTAL SETUP

**Pretrained Network.** We assume that we have gathered prior knowledge on unrelated visual tasks. This knowledge is modeled by a deep convolutional network, trained on a large-scale dataset. In our experiments, we choose to use a ResNet-18 (He et al., 2016) *pre-trained* on the Places365-Standard subset of Places365 (Zhou et al., 2017). We refer to this subset as Places. This subset contains around 1.8 million images across 365 classes. The classes are outdoor and indoor scenes. We select this dataset for its large scale, diversity of content and different nature than other popular datasets like CUB-200-2011 (see below). Images are resampled to $224 \times 224$ pixels for training. We choose ResNet-18 as the architecture of the pre-trained model as it is a powerful network that is also used in other few-shot learning studies (Chen et al., 2019; Dvornik et al., 2019), which helps in comparisons. We make no assumption on the pre-training of the network. We do not access either the pre-training process or the dataset. We rather use a publicly available converged model that has been trained with a fully-connected layer as a classifier, as assumed in section 3.

**Datasets.** We apply our method to two standard datasets in few-shot learning. The first is CUB-200-2011 (Wah et al., 2011), referred to as CUB, originally meant for fine-grained classification, and subsequently introduced to few-shot learning by Hilliard et al. (2018). This dataset contains 11,788 images of birds across 200 classes corresponding to different species. We use the split proposed by Ye et al. (2018), where 100 classes are used as base classes and the remaining 100 as novel, out of which 50 for validation and 50 for testing. CUB images are cropped using bounding box annotations and resampled to $224 \times 224$.

| | NOVEL: $k'=1$ | | | | NOVEL: $k'=5$ | | | |
|---|---|---|---|---|---|---|---|---|
| Attention | | ✓ | | ✓ | | ✓ | | ✓ |
| Adaptation | | | ✓ | ✓ | | | ✓ | ✓ |
| BASE | PLACES | | | | | | | |
| $k=0$ | 38.80±0.24 | 39.69±0.24 | 39.76±0.24 | 40.79±0.24 | 55.09±0.24 | 56.95±0.23 | 63.29±0.24 | 64.27±0.23 |
| $k=1$ | 40.50±0.23 | 41.74±0.24 | 41.11±0.24 | 42.23±0.24 | 57.25±0.22 | 58.89±0.23 | 65.42±0.23 | 66.78±0.23 |
| $k=5$ | 56.47±0.28 | 57.16±0.29 | 56.69±0.29 | 57.32±0.29 | 74.27±0.23 | 74.95±0.23 | 75.82±0.23 | 76.32±0.23 |
| ALL | 80.68±0.27 | 80.48±0.27 | 80.68±0.27 | 80.56±0.27 | 90.38±0.16 | 90.33±0.16 | 91.22±0.15 | 91.17±0.15 |
| BASE | RANDOMLY INITIALIZED | | | | | | | |
| $k=1$ | 31.65±0.19 | - | 31.37±0.19 | - | 39.45±0.20 | - | 42.70±0.21 | - |
| $k=5$ | 40.52±0.25 | - | 40.50±0.26 | - | 52.94±0.25 | - | 53.45±0.25 | - |
| ALL | 71.78±0.30 | - | 71.77±0.30 | - | 85.60±0.18 | - | 85.96±0.19 | - |
| Baseline++ | 67.02±0.90 | - | - | - | 83.58±0.54 | - | - | - |
| ProtoNet | 71.88±0.91 | - | - | - | 87.42±0.48 | - | - | - |
| Ensemble | 68.77±0.71 | - | - | - | 84.62±0.44 | - | - | - |

Table 1: *Average 5-way $k'$-shot novel class accuracy on CUB.* We use ResNet-18 either pre-trained on Places or we train it from scratch on $k$ base class examples. ProtoNet (Snell et al., 2017) is as reported by Chen et al. (2019). For ensemble (Dvornik et al., 2019), we report the distilled model from an ensemble of 20. Baselines to be compared only to randomly initialized with $k=$ ALL.

The second dataset is *mini*ImageNet (Vinyals et al., 2016), a subset of the ImageNet ILSVRC-12 (Russakovsky et al., 2014) containing 100 classes with 600 images per class. Following the split from Ravi & Larochelle (2017), 64 classes are used as base classes and 36 as novel, out of which 16 for validation and 24 for testing. Originally, *mini*ImageNet images have been down-sampled from the ImageNet resolution to $84\times84$. In this work, similarily to Chen et al. (2019); Dvornik et al. (2019), we resample to $224\times224$ instead, which is consistent with the choice of pre-trained network.

Contrary to CUB, *mini*ImageNet has some non-negligible overlap with Places. Some classes or even objects appear in both datasets. To better satisfy our assumption of domain gap, we remove the most problematic overlapping classes from *mini*ImageNet. As detailed in Appendix A, we remove 3 base classes, 1 validation class and 2 novel classes. We refer to this pruned dataset as *modified* mini*ImageNet*. For the sake of comparison and because this overlap can happen in practice, we also experiment on the original *mini*ImageNet, as reported in Appendix B.

**Evaluation protocol.** To adapt to our few-shot version of few-shot learning, we randomly keep only $k$ images per base class. We experiment with $k \in \{0, 1, 5\}$ and $k \in \{0, 20, 50\}$ respectively for the CUB and *mini*ImageNet. For novel classes, we use the standard setting $k' \in \{1, 5\}$. We generate a few-shot task on novel classes by selecting a support set $X'$. In particular, we sample $c'$ classes from the validation or test set and from each class we sample $k'$ images. In all experiments, $c' = 5$, *i.e.* 5-way classification. For each task we additionally sample 30 novel class images per class, to use as queries for evaluation. We report *average accuracy* and *95% confidence interval* over 5,000 tasks for each experiment. The base class training set $X$ contains $k$ examples per class for each base class in $C$. Each experiment can be seen as a few-shot classification task on few base class examples.

**Baselines.** We evaluate experiments with the network being either pre-trained on Places or randomly initialized. In both cases, we report measurements for different number $k$ of examples per base class, as well as *all* examples in $X$. In the latter case (randomly initialized), we do not use the option $k = 0$ because then there would be no reasonable representation to adapt or to perform inference on, given a few-shot task on novel classes. In all cases, we compare to the baselines of using no adaptation and no spatial attention. When learning from scratch, spatial attention is not applied as we do not have access to the pre-trained classifier. In the case of random initialization, and using all examples in $X$, we compare to Baseline++ (Chen et al., 2019) and prototypical networks (Snell et al., 2017), as reported in the benchmark by Chen et al. (2019), as well as *category traversal* (CTM) (Li et al., 2019a) and *ensembles* (Dvornik et al., 2019), all using ResNet-18. They can only be compared to our randomly initialized baseline when using base training on all data.

**Implementation details.** At base training, we use stochastic gradient descent with Nesterov momentum with mini-batches of size 200. At adaptation, we perform a maximum of 60 iterations over the support examples using Adam optimizer with fixed learning rate. In both cases, the learning rate,

|  | NOVEL: $k' = 1$ | | | | NOVEL: $k' = 5$ | | | |
|---|---|---|---|---|---|---|---|---|
| Attention | ✓ | | ✓ | | ✓ | | ✓ | |
| Adaptation | | ✓ | ✓ | | | ✓ | ✓ | |
| BASE | PLACES | | | | | | | |
| $k = 0$ | 61.66±0.30 | 63.36±0.29 | 62.09±0.30 | 63.56±0.30 | 78.86±0.22 | 80.15±0.22 | 80.38±0.22 | 81.05±0.22 |
| $k = 20$ | 62.95±0.29 | 63.15±0.28 | 63.11±0.29 | 63.33±0.29 | 78.41±0.21 | 78.53±0.21 | 79.67±0.21 | 79.82±0.21 |
| $k = 50$ | 65.07±0.29 | 65.10±0.29 | 65.18±0.29 | 65.24±0.29 | 79.94±0.20 | 79.99±0.20 | 80.88±0.20 | 80.96±0.20 |
| ALL | 66.20±0.29 | 65.94±0.29 | 66.23±0.29 | 66.06±0.29 | 80.37±0.21 | 80.24±0.21 | 81.56±0.20 | 81.50±0.20 |
| BASE | RANDOMLY INITIALIZED | | | | | | | |
| $k = 20$ | 33.43±0.21 | - | 33.35±0.21 | - | 43.83±0.21 | - | 44.21±0.21 | - |
| $k = 50$ | 41.03±0.24 | - | 41.05±0.24 | - | 54.68±0.22 | - | 54.92±0.22 | - |
| ALL | 55.99±0.28 | - | 56.13±0.28 | - | 72.43±0.22 | - | 73.10±0.21 | - |

Table 2: *Average 5-way $k'$-shot novel class accuracy on modified* mini*ImageNet.* We use ResNet-18 either pre-trained on Places or we train it from scratch on $k$ base class examples. Baselines only shown in Appendix B on the original *mini*ImageNet.

schedule if any and number of iterations are determined on the validation set. The temperature used by (7) for the computation of the entropy is fixed per dataset, again on the validation set. In particular, we use $T = 100$ and $T = 2.6$ respectively for CUB and modified *mini*ImageNet.

## 5.2 RESULTS

We present results in Tables 1 and 2 respectively for CUB and modified *mini*ImageNet.

**Effect of base training.** For fine-grained few-shot classification (CUB), base training is extremely important in adapting to the new domain, improving the baseline 1-shot accuracy by more than 40% with no adaptation and no spatial attention. On object classification in general (modified *mini*ImageNet), it is less important, improving by 4.5%. It is the first time that experiments are conducted on just a small subset of the base class training set. It is interesting that 50 examples per class are bringing nearly the same improvement as all examples, *i.e.* hundreds per class.

**Effect of (novel class) adaptation.** Fine-tuning the network on $k'$ novel class examples per class, even fewer than $k$ in the case of base classes, comes with the risk of over-fitting. We still show that a small further improvement is possible with a small learning rate. The improvement is more significant when $k$ is low, in which case, more adaptation of the embedding network to the novel class domain is needed. In the extreme case of CUB dataset without base training, adaptating on only the 25 images of the 5-way 5-shot tasks brings an improvement of 8.20%.

**Effect of spatial attention.** Spatial attention allows focusing on the most discriminative parts of the input, which is more beneficial when fewer examples are available. The extreme case is having no base class images and only one image per novel class. In this case, most improvement comes on modified *mini*ImageNet without base training, where spatial attention improves 5-way 5-shot classification accuracy by 1.5% after adaptation. The attention maps appear to be domain independent as they improve CUB accuracy even when no images from the bird domain have been seen ($k = 0$).

## 6 CONCLUSION

In this paper we address the problem of few-shot learning when even base classes images are limited in number. To address it, we use a pre-trained network on a large-scale dataset and a very simple spatial attention mechanism that does not require any training on the base or novel classes. We consider two few-shot learning datasets: CUB and *mini*ImageNet, with different domain gaps to our prior dataset Places. Our findings indicate that even when the domain gap is large between the dataset used for pre-training and the base/novel class domains, it is still possible to get significant benefit from base class training even with a few examples, which is very important as it reduces the need for supervision. The gain from spatial attention is more pronounced in this case.

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

## A  DETAILS ON REMOVING DATASET OVERLAPS

We study the overlap between Places and *mini*ImageNet by measuring, for each *mini*ImageNet class, what is the most frequent prediction among Places classes by the pre-trained Resnet-18 classifier and what proportion of examples are classified in this class. Ranking *mini*ImageNet classes by that proportion, we check class names on both datasets and manually inspect examples in the top-ranking classes. We chose to remove only the most clearly overlapping classes, that is, identical classes and classes of objects contained in images of a Places class. For instance, most images from the bus station - indoor class of Places contain a bus, so we chose to remove the school bus class from *mini*ImageNet. Table A lists the classes removed from *mini*ImageNet in this way.

| *mini*ImageNet split | *mini*ImageNet class | Places class with overlap |
|---|---|---|
| TRAIN | carousel | carousel |
| TRAIN | slot | amusement arcade |
| TRAIN | cliff | cliff |
| VALIDATION | coral reef | underwater - ocean deep |
| TEST | school bus | bus station - indoor |
| TEST | bookshop | bookstore |

Table 3: Classes removed from *mini*ImageNet to form the *modified* mini*ImageNet* dataset and the corresponding overlapping Places classes.

## B  ORIGINAL *mini*IMAGENET RESULTS

We also run our experiments on the original *mini*ImageNet dataset which partially overlaps our pre-training dataset Places. We use $T = 2.4$ for the temperature in (7). The remaining setup as for modified *mini*ImageNet. Results are shown in Table 4.

| | NOVEL: $k' = 1$ | | | | NOVEL: $k' = 5$ | | | |
|---|---|---|---|---|---|---|---|---|
| Attention | | ✓ | | ✓ | | ✓ | | ✓ |
| Adaptation | | | ✓ | ✓ | | | ✓ | ✓ |
| BASE | PLACES | | | | | | | |
| $k = 0$ | 65.80±0.31 | 67.56±0.31 | 66.41±0.32 | 67.96±0.31 | 81.90±0.23 | 83.00±0.22 | 83.45±0.22 | 84.09±0.22 |
| $k = 20$ | 66.98±0.29 | 67.63±0.29 | 67.32±0.29 | 67.80±0.29 | 81.44±0.21 | 81.82±0.21 | 82.56±0.21 | 82.92±0.21 |
| $k = 50$ | 69.11±0.29 | 69.17±0.29 | 69.22±0.29 | 69.30±0.29 | 83.14±0.20 | 83.25±0.20 | 83.97±0.19 | 84.10±0.19 |
| ALL | 69.71±0.29 | 69.81±0.29 | 69.70±0.29 | 70.00±0.29 | 83.31±0.19 | 83.25±0.19 | 84.20±0.19 | 84.24±0.19 |
| BASE | RANDOMLY INITIALIZED | | | | | | | |
| $k = 20$ | 37.75±0.23 | - | 37.74±0.23 | - | 49.13±0.23 | - | 49.67±0.23 | - |
| $k = 50$ | 42.79±0.23 | - | 42.79±0.23 | - | 57.18±0.23 | - | 57.68±0.23 | - |
| ALL | 59.68±0.27 | - | 59.66±0.27 | - | 75.42±0.20 | - | 75.95±0.20 | - |
| Baseline++ | 51.87±0.77 | - | - | - | 75.68±0.63 | - | - | - |
| ProtoNet | 54.16±0.82 | - | - | - | 73.68±0.65 | - | - | - |
| Ensemble | 63.06±0.63 | - | - | - | 80.63±0.43 | - | - | - |
| CTM | 64.12±0.55 | - | - | - | 80.51±0.13 | - | - | - |

Table 4: *Average 5-way $k'$-shot novel class accuracy on* mini*ImageNet.* We use ResNet-18 either pre-trained on Places or we train it from scratch on $k$ base class examples. ProtoNet (Snell et al., 2017) is as reported by Chen et al. (2019). CTM refers to the data-augmented version of Li et al. (2019a). For ensemble (Dvornik et al., 2019), we use the distilled model from an ensemble of 20. Baselines to be compared only to randomly initialized with $k = $ ALL.

Compared to the results of modified *mini*ImageNet (Table 2), performances are nearly uniformly increased by 3-4% and conclusions remain the same. The increase in performance is due to having more training data, as well as putting back easily classified classes in the test dataset. Observe that, unlike CUB (*cf.* Table 1), CTM (Li et al., 2019a) and ensembles (Dvornik et al., 2019) perform better than our randomly initialized baseline . Our objective is not to improve the state of the art of the standard few-shot setup , but rather to study the new problem using a network pre-trained on a large-scale dataset. In this respect, our simple baseline better facilitates future research.

