# OpenReview forum: "Few-Shot Few-Shot Learning and the role of Spatial Attention"
_ICLR.cc/2020/Conference — Reject_

### Official Review · AnonReviewer1 · 2019-10-21
**Official Blind Review #1**

**Rating:** 1

**Review:**

The paper introduces a problem “few-shot few-shot learning” that aims to firstly transfer prior knowledge from one domain to the domain where the base training tasks reside, and then train a few-shot learning model on training tasks and apply it to novel test tasks. The two “few-shot” in the name refers to base training tasks and novel test tasks. In their algorithm, they use a model pre-trained on another dataset as the prior knowledge and fine-tune it on training tasks. During the test, they use the weighted average of samples’ representations per class as the prototype of each class, where the weight is large for samples with more discriminative prediction over pre-trained domain’s classes. Afterward, classification is reduced to finding the nearest neighbor among the class prototypes. Some experiments show that the pre-trained model can improve few-shot classification accuracy.

My major concerns:

1) They try to propose a new problem, but their description shows that the problem is exactly the same as what most “few-shot learning” works aim to solve: use a pre-trained model, train a meta-learner on few-shot training tasks, and apply it to novel test tasks.

2) The algorithm does not have any important contributions comparing to existing ones: they define a prototype per class based on the pre-trained model and apply the nearest neighbor classification. The so-called “prototypical classifier” is actually the nearest neighbor classifier since no prototypical network structure is learned in the proposed method.

3) I would not call the weighted average as “attention” because it is not: the weight in attention is computed by a module with learnable parameters, while the weight in this paper is computed by the entropy of a pre-defined model’s output prediction.

4) The “spatial attention” only makes sense when the pre-trained domain’s classes can describe the main concepts appearing in the images of novel classes. This assumption is too strong since it requires class-level (rather than lower-level) relationships.

5) The base training is not necessary in the algorithm: it is used to only fine-tuning theta and W. As the author said in the beginning of Section 4.1, they can directly solve novel tasks based on the pre-trained model.

6) The experiments show that the pre-trained model is helpful in few-shot learning, which is a known fact.

7) The writing of this paper is very poor: a lot of typos and grammar errors, inconsistency between narratives, abuse of notations, wrong equation reference, even missing punctuations. They make the paper hard to understand.

-------------

Update:

Thanks for the authors' rebuttal! After reading their rebuttal, I still have main concerns about the novelty of the problem and the writing quality. The proposed method tends to be incremental.

**Experience Assessment:**

I have published one or two papers in this area.

**Review Assessment: Checking Correctness Of Derivations And Theory:**

I carefully checked the derivations and theory.

**Review Assessment: Checking Correctness Of Experiments:**

I carefully checked the experiments.

**Review Assessment: Thoroughness In Paper Reading:**

I read the paper thoroughly.

---

> ### Author Response · Authors · 2019-11-15
> **Response to Reviewer #1**
>
> Most concerns appear to be due to a very poor understanding of our work by the Reviewer, which is hard to explain for someone having published in this area and read the paper thoroughly. We do our best to explain below.
>
>
> C1: "They try to propose a new problem, but their description shows that the problem is exactly the same as what most “few-shot learning” works aim to solve: use a pre-trained model, train a meta-learner on few-shot training tasks, and apply it to novel test tasks."
>
> A1: In summary: as explained in the 4th paragraph of the introduction, our problem differs from standard few-shot learning in that a large scale dataset from another domain is available, while in-domain base class data is lacking. This problem is more realistic than the common few-shot setting, as the other two reviewers agree.
>
>
> C2: "The algorithm does not have any important contributions comparing to existing ones: they define a prototype per class based on the pre-trained model and apply the nearest neighbor classification. The so-called “prototypical classifier” is actually the nearest neighbor classifier since no prototypical network structure is learned in the proposed method."
>
> A2: As we explain in section 2/"Contribution", our contribution is to introduce this new variant of few-shot learning and study the effects of spatial attention and adaptation in this new setting. We do not claim anywhere that the prototype classifier is our contribution. On the contrary, we discuss it in section 2/"Prototypes" as background. In section 2/"Related work", we also explain how prototypical networks (Snell et al., 2017) use this classifier in a meta-learning setup.
>
> We use this prototype classifier in the adaptation stage where at each iteration, the prototypes of the novel classes are computed with the (updated) features of the support examples. We use standard cross-entropy loss on the output of a cosine classifier (2) having the prototypes as class weights, as described in section 4.2. This may not be a standard choice, but it is not a claimed contribution either. At inference, the prediction is indeed the nearest prototype as in (Snell et al., 2017).
>
>
> C3: "I would not call the weighted average as "attention" because it is not: the weight in attention is computed by a module with learnable parameters, while the weight in this paper is computed by the entropy of a pre-defined model’s output prediction."
>
> A3: As a matter of terminology, visual attention has existed in computer vision long before being computed by a learnable module. A very well-known example is Itti, Koch and Niebur, A model of saliency-based visual attention for rapid scene analysis, PAMI 1998.
>
>
> C4: "The "spatial attention" only makes sense when the pre-trained domain’s classes can describe the main concepts appearing in the images of novel classes. This assumption is too strong since it requires class-level (rather than lower-level) relationships."
>
> A4: As shown in Figure 1 and as demonstrated in the results, our spatial attention mechanism can indeed generalize to novel classes, which is remarkable for its simplicity and absence of learnable parameters. As we discuss, an interpretation is that "uncertainty over a large number of such classes may express anything unknown like background." Reviewer #2 appears to agree on the validity of this argument.
>
>
> C5: "The base training is not necessary in the algorithm: it is used to only fine-tuning theta and W. As the author said in the beginning of Section 4.1, they can directly solve novel tasks based on the pre-trained model."
>
> A5: Certainly we can solve new tasks, but is that good enough? On CUB for instance (Table 1), k=5 can be up to 18% better than k=0; k=ALL can be more than 40% better.
>
>
> C6: "The experiments show that the pre-trained model is helpful in few-shot learning, which is a known fact."
>
> A6: Pre-training on a large-scale dataset from another domain has not been studied in the context of few-shot learning. This comment is apparently due to the poor understanding of the problem, as addressed in A1.
>
>
> C7: "The writing of this paper is very poor: a lot of typos and grammar errors, inconsistency between narratives, abuse of notations, wrong equation reference, even missing punctuations. They make the paper hard to understand."
>
> A7: We are always open to constructive criticism. A comment as severe as this would deserve at least some concrete examples for each kind of writing error.

---

### Official Review · AnonReviewer2 · 2019-10-23
**Official Blind Review #2**

**Rating:** 3

**Review:**

This paper proposed a new realistic setting for few-shot learning that we can obtain representations from a pre-trained model trained on a large-scale dataset, but cannot access its training details. Also, there may be a large domain shift between the dataset of the pre-trained model and our dataset. For the pre-trained model, they will not only use its weights but also use it to generate a spatial attention map and help the model focuses on objects of images. Back to the standard few-shot classification problem, they will first adapt the model with base class samples and then adapt to novel classes.

The proposed new setting is very meaningful since we already have many powerful pre-trained models and why not exploit its usage for few-shot learning problems. However, I doubt the novelty and effectiveness of the attention way used in the paper. The attention module helps the model focuses on the objects not the background, which is absolutely correct. But there are already some relevant studies in the missing reference Large-Scale Long-Tailed Recognition in an Open World, CVPR2019. Also, from the results, the significant improvements come from the weights of the pre-trained model but not the attention used. Is the attention way used in the paper a good way to exploit the pre-trained model for few-shot classification problems?

Also, I am curious about the dense classification used in the adaptation phase. Will it achieve similar performance with finetuning using just standard loss?

Btw, according to the formatting instructions, the abstract should be limited in one paragraph.

=========================================================
After Rebuttal:

I thank the author for the response.

I do see there are differences in the way of generating attention masks between the proposed work and (Liu et al.). But the improvements from the attention module is not significant, especially when using all base data.

I keep my original scores.

**Experience Assessment:**

I have published one or two papers in this area.

**Review Assessment: Checking Correctness Of Derivations And Theory:**

N/A

**Review Assessment: Checking Correctness Of Experiments:**

I carefully checked the experiments.

**Review Assessment: Thoroughness In Paper Reading:**

I read the paper thoroughly.

---

> ### Author Response · Authors · 2019-11-15
> **Response to Reviewer #2**
>
> Thank you for your review. Please find our response below.
>
> C1: "However, I doubt the novelty and effectiveness of the attention way used in the paper. The attention module helps the model focuses on the objects not the background, which is absolutely correct. But there are already some relevant studies in the missing reference Large-Scale Long-Tailed Recognition in an Open World, CVPR2019."
>
> A1: Thank you for pointing out this related work, which we shall discuss. Liu et al. also use an attention mechanism in their work. However, the attention is computed from the feature maps of the embedding network trained on in-domain classes. In our work, because in-domain base classes images might not be available we propose an attention mechanism that is as generic as possible. Their ablation study shows that using a spatial attention mechanism can improve few-shot accuracy by a small margin of not more than 1%, which is consistent with our findings.
>
>
> C2: "Also, from the results, the significant improvements come from the weights of the pre-trained model but not the attention used."
>
> A2: Of course, a pre-trained model performs a lot better than a model trained from scratch on the base class data of standard few-shot learning benchmarks, even when the pre-training and few-shot domains are very different (Places and CUB). This is exactly our motivation to study a practical setting that is completely overseen in all work on few-shot learning so far. That said, the choice of pre-trained model is not a method to be compared to others but rather what defines the problem (in all tables for instance, we explicitly say "Baselines to be compared only to randomly initialized with k=ALL"). In this problem, the gains coming from attention/adaptation are in general consistent across all experiments and can be up to 8% as discussed in A3 to Reviewer #3. These findings are interesting for a problem that is studied for the first time.
>
>
> C3: "Also, I am curious about the dense classification used in the adaptation phase. Will it achieve similar performance with fine-tuning using just standard loss?"
>
> A3: As described in section 4.2, during the adaptation phase, we use the standard cosine classifier and the loss function is standard cross-entropy. Dense classification is only used during base class adaptation, similarly to (Lifchitz et al., 2019). We also experimented with dense classification at stage 2, which was inferior. We can discuss that.
>
>
> C4: "Btw, according to the formatting instructions, the abstract should be limited in one paragraph."
>
> A4: Fixed.

---

### Official Review · AnonReviewer3 · 2019-10-23
**Official Blind Review #3**

**Rating:** 3

**Review:**

A new task is suggested, similarly to FSL the test is done in an episodic manner of k-shot 5-way, but the number of samples for base classes is also limited. The model is potentially pre-trained on a large scale dataset from another domain. The suggested method is applying spatial attention according to entropy criteria (or certainty) of the original classifier (from a different domain).


I think the suggested task is important and more realistic than the usual FSL benchmarks. I would modify it so instead of discarding mini-imagenet classes that are overlapping with Places I would discard the problematic Places classes. This way it will be easier to compare to standard FSL. Also, I don’t understand why for CUB the benchmarks includes k={0,1,5} while for mini-imagenet it is k={0,20,50}, obviously k={0,1,5} are more interesting.

As for the suggested method, I find it hard to judge since there are no strong baselines to compare against. Also, the ablation study of removing the attention and/or adaptation doesn’t result in a definitive conclusion.


Update:
While your comments do weaken some of my concerns, I'm afraid it is not enough for changing my previous rating. I think being more careful about the benchmark definition with regards to train/test overlap and comparing to stronger baselines will help improve the paper for future submissions.

**Experience Assessment:**

I have published one or two papers in this area.

**Review Assessment: Checking Correctness Of Derivations And Theory:**

I assessed the sensibility of the derivations and theory.

**Review Assessment: Checking Correctness Of Experiments:**

I assessed the sensibility of the experiments.

**Review Assessment: Thoroughness In Paper Reading:**

I read the paper at least twice and used my best judgement in assessing the paper.

---

> ### Author Response · Authors · 2019-11-15
> **Response to Reviewer #3**
>
> Thank you for your review. Please find our response below.
>
> C1: "I would modify it so instead of discarding miniImageNet classes that are overlapping with Places I would discard the problematic Places classes. This way it will be easier to compare to standard FSL."
>
> A1: In Appendix B, we show results on the full miniImageNet dataset. Compared to the modified version, all results are increased nearly uniformly regardless of the initialization being random or pretrained. Therefore, conclusions made with the modified miniImageNet hold when there is some overlap between the pre-training dataset and the few-shot dataset. Removing the overlapping classes from Places before pre-training is something that we explicitly exclude in the definition of the problem: "we do not have access to its training process or data", as stated at several places. This choice stems from a very practical consideration: it allows for large networks pre-trained on large-scale data, which can be used off-the-shelf without repeating the process in every paper. This makes it harder to control overlap, which is however compensated by limiting the amount of base class data.
>
>
> C2: "Also, I don't understand why for CUB the benchmarks includes k={0,1,5} while for miniImageNet it is k={0,20,50}, obviously k={0,1,5} are more interesting."
>
> A2: Contrary to CUB where k=1 is already bringing noteworthy improvements over k=0, base training with k<20 images on miniImageNet per class does not bring significant improvement. One interpretation is that CUB, being a fine-grained dataset, has low variety of visual content such that few examples are enough to adapt the pre-trained model to the new domain. Another interpretation is that due to the larger domain gap between Places and CUB, few examples can bring significant improvement. We shall discuss. We can of course add a few more measurements as well.
>
>
> C3: "As for the suggested method, I find it hard to judge since there are no strong baselines to compare against."
>
> A3: We are comparing to recent methods that are using the same embedding network and data for fair comparison. The baseline we provide (base class training with dense classification) is simple but has state of the art performance in the classic few-shot setup. On CUB, it is only outperformed slightly by the recent implementation of prototypical network of Chen et. al. On miniImageNet (Appendix B) our baseline outperforms the same implementation by a large margin. The two models that outperform our baseline on miniImageNet (Ensemble, CTM) are more complex and applying them in our setup is not straightforward (for instance, initializing an ensemble out of an off-the shelf network). Since this problem is studied for the first time, easy reproduction is important. Besides, on CUB, Ensemble is outperformed by our baseline and CTM is not available. Overall, considering performance and simplicity, we have found our baseline the best choice. We are already discussing this at the end of Appendix B.
>
>
> C4: "Also, the ablation study of removing the attention and/or adaptation doesn’t result in a definitive conclusion."
>
> A4: Spatial attention improves few-shot classification when few examples of the base classes are available (k is small), which is consistent across datasets. When k is larger, the gain from spatial attention is lower but is still consistent. When k=ALL, spatial attention does not bring any accuracy improvement but does not degrades it significantly either (greatest loss is by 0.3%), making it a safe choice especially for small k. Improvements of adaptation are consistent everywhere. The gain is impressive (up to 8%) on CUB with few base classes (k=0 or k=1). As we discuss, this is very important because labeled base class data may not be available.

---

### Decision · Program_Chairs · 2019-12-19

**Decision:**

Reject

**Comment:**

This paper tackles the interesting problem of meta-learning in problem spaces where training "tasks" are scarce.  Two criticisms that seems to shared across reviewers are that (i) it is debatable how "novel" the space of meta learning with "few" tasks is, especially since there aren't established standard for how many training tasks should be available, and (ii) the paper could use more comparisons with baseline methods and ablations to understand the contributions.  As an AC, I down-weight criticism (i) because I don't feel the paper has to be creating a new problem definition; it's acceptable to make advances within an existing space.  However, criticism (ii) seems to remain.  After conferring with reviewers it seems that the rebuttal was not strong enough to significantly alter the reviewer's opinions on this issue, and so the paper does not have enough support to justify acceptance.  The paper certainly addresses interesting issues, and I look forward to seeing a revised/improved version at another venue.